# PKCiota Inhibits the Ferroptosis of Esophageal Cancer Cells via Suppressing USP14-Mediated Autophagic Degradation of GPX4

**DOI:** 10.3390/antiox13010114

**Published:** 2024-01-17

**Authors:** Hao Tao, Sheng-Jie Song, Ze-Wen Fan, Wen-Ting Li, Xin Jin, Wen Jiang, Jie Bai, Zhi-Zhou Shi

**Affiliations:** 1Medical School, Kunming University of Science and Technology, Kunming 650500, China; sunxiatianchen@126.com (H.T.); songshengjie1998@163.com (S.-J.S.); 15879764290@163.com (Z.-W.F.); liwenting2035@163.com (W.-T.L.); jinxin8833@163.com (X.J.); jiebai662001@126.com (J.B.); 2Department of Thoracic Surgery, The First People’s Hospital of Yunnan Province, The Affiliated Hospital of Kunming University of Science and Technology, Kunming 650000, China; coolgzy1314@126.com

**Keywords:** PKCiota, GPX4, USP14, ferroptosis, esophageal squamous cell carcinoma

## Abstract

Esophageal squamous cell carcinoma (ESCC) is one of the most frequent malignant tumors, and the mechanisms underlying the anti-ferroptosis of esophageal cancer cells are still largely unclear. This study aims to explore the roles of amplified protein kinase C iota (PKCiota) in the ferroptosis of ESCC cells. Cell viability, colony formation, MDA assay, Western blotting, co-IP, PLA, and RNA-seq technologies are used to reveal the roles and mechanisms underlying the PKCiota-induced resistance of ESCC cells to ferroptosis. We showed here that PKCiota was amplified and overexpressed in ESCC and decreased during RSL3-induced ferroptosis of ESCC cells. PKCiota interacted with GPX4 and the deubiquitinase USP14 and improved the protein stability of GPX4 by suppressing the USP14-mediated autophagy–lysosomal degradation pathway. PKCiota was negatively regulated by miR-145-5p, which decreased in esophageal cancer, and also regulated by USP14 and GPX4 by a positive feedback loop. PKCiota silencing and miR-145-5p overexpression suppressed tumor growth of ESCC cells in vivo, respectively; even a combination of silencing PKCiota and RSL3 treatment showed more vital suppressive roles on tumor growth than silencing PKCiota alone. Both PKCiota silencing and miR-145-5p overexpression sensitized ESCC cells to RSL3-induced ferroptosis. These results unveiled that amplified and overexpressed PKCiota induced the resistance of ESCC cells to ferroptosis by suppressing the USP14-mediated autophagic degradation of GPX4. Patients with PKCiota/USP14/GPX4 pathway activation might be sensitive to GPX4-targeted ferroptosis-based therapy.

## 1. Introduction

Esophageal squamous cell carcinoma (ESCC) is one of the highest-incidence malignant tumors in the world and ranks fifth in cancer death in China [1]. In the past decades, methods of early detection and therapy have been developed; however, the prognosis of ESCC patients is still unfavorable [2]. Thus, revealing the mechanisms underlying esophageal carcinogenesis and developing new diagnostic and prognostic methods for ESCC patients are urgent and essential.

Ferroptosis is defined as a newly discovered and iron-dependent non-apoptotic cell death with characteristics of lethal lipid peroxides [3]. Inducing ferroptosis could reverse drug resistance, and targeting ferroptosis will be a promising method for cancer therapy [4,5]. Glutathione peroxidase 4 (GPX4) is a crucial regulator of ferroptosis, which could eliminate toxic lipid hydroperoxides (LOOH) and protect cells from oxidative stress. Inhibition of GPX4 promotes ferroptosis by inducing lipid-reactive oxygen species (ROS) [6]. Previous studies have reported that the overexpression of GPX4 was a prognostic factor in ESCC, and GPX4 could be regulated by circPVT1, DNAJB6, and 5-aminolevulinic acid (5-ALA) in ESCC cells [7,8,9]. Nevertheless, in ESCC, the regulatory mechanism of anti-ferroptosis of cancer cells is largely unclear.

Protein kinase C (PKC) consists of three subfamilies of phospholipid-dependent serine/threonine kinases, including classic (PKCα, PKCβI, PKCβII, and PKCγ), non-classic (PKCδ, PKCε, PKCη, and PKCθ) and atypical (PKCζ, PKCι, and PKCλ) isozymes [10,11]. Previous studies reported that PKC played dual roles in the ferroptosis of cancer cells. On the one hand, PKC promotes cancer cell ferroptosis. PRKCQ (PKCζ) could phosphorylate HPCAL1 on Thr149, promote the autophagic degradation of CDH2, and then increase cancer cells’ susceptibility to ferroptosis [12]. PKCβII could promote the phosphorylation and activation of ACSL4, amplify lipid peroxidation, and induce ferroptosis, and suppressing the PKCβII–ACSL4 signaling pathway blocked ferroptosis [13]. The broad-spectrum inhibitor of PKC (Bisindolylmaleimide I) and inhibitor of PKCα and PKCβ (Gö6976) could reduce the Erastin-induced ferroptosis of rhabdomyosarcoma cells [14]. On the other hand, PKC induces resistance to cancer cell ferroptosis. PKC (especially PKCε and PKCα) could promote the phosphorylation of HSPB1 and protect cancer cells from ferroptosis [15]. However, up to now, the roles of other PKCs, including PKCι, in regulating the ferroptosis of cancer cells are still absolutely unknown.

PKCι (PKCiota) is amplified and overexpressed in many cancers and plays oncogenic roles in carcinogenesis. In ovarian cancer, PKCiota was amplified and overexpressed, and depletion of PKCiota induced apoptosis of ovarian cancer cells [16]. In pancreatic cancer, PKCiota upregulated sp1, promoted sp1 to bind with the promoter of YAP1 and transactivated YAP1 expression, and finally promoted pancreatic carcinogenesis [17]. In glioblastoma (GBM), knockdown or inhibition of PKCiota significantly suppressed tumor growth and prolonged survival time in animal models, PKCiota and SRC signaling were reciprocally related, and a combination of the PKCiota inhibitor auranofin and an SRC signaling inhibitor showed prolonged survival [18]. Significantly, PKCiota could promote the immune-suppressive microenvironment in pancreatic cancer by elevating PDL1 expression, and a combination of a PDL1 inhibitor and a PKCiota inhibitor could remarkably enhance the cytotoxicity of NK92 against PDAC cells [19]. Copy number gains were the critical reason for PKCiota overexpression in cancers [20].

We previously reported that PKCiota was amplified and overexpressed in ESCC, and both its amplification and overexpression were correlated with lymph node metastasis in ESCC patients [21]. PKCiota induced resistance to anoikis and promoted the metastasis of ESCC via inhibiting the proteasome-dependent degradation of SKP2 [22]. PKCiota also accelerated the G1/S transition and promoted ESCC cell proliferation by interacting and phosphorylating CDK1 [23]. In ESCC, PKCiota could be interacted with and positively regulated by p62 [24]. However, whether PKCiota handles the ferroptosis of tumor cells is still unknown.

Our study revealed that PKCiota declined during RSL3-caused ferroptosis of ESCC cells and silenced PKCiota-sensitized ESCC cells to RSL3-induced ferroptosis. We further investigated the mechanisms of PKCiota in ESCC cell ferroptosis.

## 2. Materials and Methods

### 2.1. Patients and Tissue Samples

Forty-eight surgically resected ESCC tissues were collected by the Department of Thoracic Surgery at the First Hospital of Yunnan Province and the Affiliated Hospital of Kunming University of Science and Technology, Kunming, China. Each patient received no therapy before surgery and signed an informed consent form. The study was conducted in accordance with the Declaration of Helsinki and approved by the Medical Ethics Committee of Kunming University of Science and Technology (KMUST-MEC-065; date of approval: 11 March 2021).

### 2.2. Cell Lines, Transfection, and Reagents

KYSE450 and KYSE510 ESCC cell lines, which were purchased from COBIOER BIOSCIENCE (Nanjing, China), were cultured using RPMI-1640 culture medium supplemented with 10% fetal bovine serum, streptomycin (100 mg/mL), and penicillin (100 U/mL) under conditions of 5% CO_2_ and 37 °C. Lipofectamine 2000 (Thermo Scientific, Carlsbad, CA, USA) was used to transfect siRNAs and vectors.

The PKCiota siRNAs, USP14 siRNA, GPX4 siRNAs, CANX siRNA, miR-145-5p mimic, NC siRNA, shPKCiota vectors, and pGCMV/EGFP/miR-145-5p vectors were synthesized by GenePharma Co., Ltd. (Shanghai, China). Information about the siRNAs is listed in Appendix A.

RSL3, Erastin, FIN56, Z-VAD-FMK, liproxstatin-1, necrostatin-1, ferrostatin-1, MG132, BafA1, IU1, and MK-2206 were obtained from Selleck Chemicals (Houston, TX, USA).

### 2.3. RNA Preparation and qRT-PCR

RNA preparation and qRT-PCR experiments followed our previous method [25]. The PCR program was as follows: (1) 50 °C for 2 min and 95 °C for 2 min; (2) 40 cycles of 95 °C for 15 s and 60 °C for 1 min. Information on primers is listed in Appendix A. The internal control was U6 and GAPDH. Experiments were performed in technical triplicates and in biological triplicates.

### 2.4. Western Blotting Assay

A Western blotting assay was carried out following our previous method [25]. Information about the antibodies is as follows: anti-PKCiota antibodies (BD Biosciences, Franklin Lakes, NJ, USA, 610175, 1:1000 dilution; Proteintech, Wuhan, China, 13883-1-AP, 1:1000 dilution), anti-GPX4 antibodies (Abcam, Cambridge, UK, ab125066, 1:2000 dilution; Proteintech, 67763-1-Ig, 1:1000 dilution), anti-USP14 antibodies (Cell signaling technology, Boston, USA, 11931S, 1:1000 dilution; Proteintech, 14517-1-AP, 1:2000 dilution), anti-p62 antibody (MBL, M162-3, 1:1000 dilution), anti-K63 linkage-specific polyubiquitin antibody (Cell Signaling technology, 5621, 1:1000 dilution), anti-p-AKT antibody (Cell Signaling technology, 4060, 1:2000 dilution), anti-AKT antibody (Cell Signaling technology, 4691, 1:1000 dilution), anti-CANX antibody (Proteintech, 10427-2-AP, 1:5000 dilution), anti-SOX2 antibody (Proteintech, 11064-1-AP, 1:1000 dilution), and anti-GAPDH antibodies (Abcam, ab8245, 1:2000 dilution; Proteintech, 10494-1-AP, 1:5000 dilution). Measurements were carried out in biological triplicates.

### 2.5. Immunoprecipitation Analysis and Proximity Ligation Assay

The interaction among PKCiota, GPX4, and USP14 was analyzed using a co-immunoprecipitation (co-IP) assay [25]. The cells were lysed using an IP lysis buffer (non-denaturing) for one hour at 4 °C. The homogeneous protein G-agarose was used to decrease non-specific binding. A rabbit PKCiota polyclonal antibody (Proteintech, 13883-1-AP), a rabbit GPX4 monoclonal antibody (Abcam, ab125066), a rabbit phosphoserine polyclonal antibody (Abcam, ab9332), and a mouse ubiquitin monoclonal antibody (Cell signaling technology, 3936S) were used for immunoprecipitation in co-IP assay. Measurements were carried out in biological triplicates.

The interaction between PKCiota and GPX4 was measured using a proximity ligation assay (PLA) using Duolink In situ Red Starter Kit (DUO92101, Sigma-Aldrich, St Louis, MO, USA). Cells were grown on glass coverslips and then fixed using 4% paraformaldehyde. After washing using PBS (containing glycine), the samples were permeabilized using PBS (containing 0.1% Triton X-100) for 20 min, then blocked and incubated using GPX4 antibody (Abcam, ab125066) and PKCiota antibody (BD Biosciences, 610175) overnight at 4 °C. Then, samples were treated using pre-diluted anti-mouse minus as well as anti-rabbit plus probes for one h at 37 °C. Treatments of 1 × ligase (37 °C for 30 min) and 1 × polymerase (37 °C for 100 min) were applied. Ultimately, Duolink In situ Mounting Medium with DAPI was used to treat the coverslips. The images were collected using the Nikon AXR Confocal Microscope (Tokyo, Japan). Measurements were carried out in biological triplicates.

### 2.6. Colony Formation and Cell Viability

Colony formation ability and cell viability were detected according to our previously described method [25]. In the colony formation assay, 10^3^ cells (per well) were seeded in a 6-well plate and cultivated for 7–10 days. At the end of this experiment, crystal violet (Sigma-Aldrich, St Louis, MO, USA) was harnessed to stain the clones, and images were captured by the Nikon Eclipse TE2000-S microscope (Japan). Measurements were carried out in biological triplicates. In the cell viability assay, 5 × 10^3^ cells (per well) were seeded in 96-well plates. The detection was performed at 48 h after PKCiota siRNAs transfection, at 24 h after RSL3 and inhibitor treatment, and at 48 h after transfection of PKCiota siRNA or miR-145-5p mimics and treatment with RSL3 (24 h). One hour before detection, 10 μL of Cell Counting Kit-8 reagent (CCK8, Dojindo Laboratories, Kumamoto, Japan) was added. Finally, the absorbance (450 nm) was detected using a plate reader (Molecular Devices, Spectra Max 190, Sunnyvale, CA, USA). Experiments were performed in technical triplicates and in biological triplicates.

### 2.7. MDA Assay

Lipid peroxidation was evaluated using a Lipid Peroxidation (MDA) assay kit (colorimetric) following the instructions of the manufacturer (Abcam, Cambridge, UK). Analysis was performed in technical triplicates and in biological triplicates.

### 2.8. Immunohistochemistry Analysis

Immunohistochemistry (IHC) analysis followed our previously described method [25]. Slides were deparaffinized, rehydrated, and incubated in a 3% hydrogen peroxide solution (10 min). Then, the slides were heated in citrate buffer (pH 6.0) at 95 °C for 25 min and cooled at room temperature for 60 min. After blocking with 10% goat serum (37 °C, 30 min), the slides were incubated with rabbit anti-PKCiota antibody (Proteintech, 13883-1-AP, 1:200 dilution) and mouse anti-GPX4 antibody (Proteintech, 67763-1-Ig, 1:1000 dilution) overnight at 4 °C followed by incubation with secondary antibody. PV-9000 and DAB kits were applied to visualize the signals.

### 2.9. Xenograft Assay

The Animal Ethics Committee of Kunming University of Science and Technology approved the animal study protocol (date of approval: 11 March 2021). Female nude mice (BALB/c, 6–8 weeks) were randomly distributed into four groups (NC and shPKCiota, n = 4 for each group; NC and miR-145-5p-OE, n = 3 for each group). Subcutaneous injection (2 × 10^6^ cells) was performed. Tumor sizes were detected weekly, and the equation (length × width^2^)/2 was applied to calculate the tumor volume. Mice were sacrificed, and cancer samples were prepared at the end of the experiments.

After the subcutaneous injection of 2 × 10^6^ shPKCiota or NC KYSE510 cells, the mice were divided into three groups (NC group, shPKCiota group, and shPKCiota + RSL3 group, n = 4/group; intraperitoneal injection with RSL3 (5 mg/kg) once per day for shPKCiota + RSL3 group). Tumor volume was detected and calculated using the above method. Mice were sacrificed, and cancer samples were prepared at the end of the experiments.

### 2.10. Databases Analysis

Genomic changes of PKCiota, expression of PKCiota and miR-145-5p, and correlation between copy number and mRNA level of PKCiota were analyzed using the databases cBioPortal (https://www.cbioportal.org, accessed on 22 December 2021), DepMap (https://depmap.org/portal/, accessed on 22 December 2021), and GEPIA (http://gepia.cancer-pku.cn/index.html, accessed on 22 December 2021) and datasets from Gene Expression Omnibus (GEO).

### 2.11. RNA-Seq Assay and Data Analysis

Global transcriptome analysis was performed in GPX4-silenced and control KYSE510 cells by Annoroad Co (Beijing, China). Differentially expressed genes (DEGs) were identified using the parameters of Fold Change < 0.5 or >2.0 and *p* < 0.05 (GPX4-silenced cells compared with control cells). KEGG pathway and gene annotation analyses based on DEGs were carried out through DAVID software (https://david.ncifcrf.gov). A Protein–Protein Interaction (PPI) network was created by the STRING website based on DEGs. MCODE and cytoHubba plugins in Cytoscape were used to analyze the module and hub genes.

### 2.12. Statistical Analyses

Statistical analysis was carried out by GraphPad Prism 9 (GraphPad, San Diego, CA, USA). Quantitative data were shown as mean ± SD and analyzed by ANOVA, χ^2^ test, and Student’s *t*-test. Overall survival time was evaluated using the Kaplan–Meier method (log-rank test). *p* < 0.05 was defined as statistical significance.

## 3. Results

### 3.1. PKCiota Was Amplified and Overexpressed in ESCC

In TCGA data, genetic changes in *PRKCI* (gene of PKCiota), including amplification and mutation, occurred in more than 15% of patients with ESCC, ovarian epithelial tumors (OV), non-small cell lung cancer (NSCLC), and cervical squamous cell carcinoma (CSCC). In ESCC, amplification (accounting for 92.86%) was the primary type (Figure 1A). By analyzing the DepMap portal database, we found that PKCiota was highly expressed in cell lines of ovarian cancer, gastric cancer, bladder cancer, and ESCC (Figure 1B). In the GEPIA database, PKCiota expression was higher in esophageal cancer tissues than in normal esophageal tissues (Figure 1C). The overexpression of PKCiota was further validated by analyzing the datasets of ESCC (GSE53624, GSE53622, and GSE20347, Figure 1D–F). Interestingly, PKCiota was also overexpressed in CSCC, and its expression was found to be higher in cancer tissues than in high-grade squamous intraepithelial lesions (HSIL) by analyzing the datasets of GSE9750 and GSE7803 (Figure 1G,H). Notably, a significant positive correlation between copy number and mRNA level of PKCiota was found in ESCC cell lines (R = 0.4442, *p* = 0.0261, Figure 1I), ESCC tissues (R = 0.7216, *p* < 0.0001, Figure 1J), and CESC tissues (R = 0.6755, *p* < 0.0001, Figure 1K). These results revealed that PKCiota was amplified and overexpressed in several types of tumors, including ESCC, and an increase in copy numbers contributed to its overexpression.

### 3.2. Knockdown of PKCiota Inhibited Cell Viability and Colony Formation Ability and Led to Ferroptosis of ESCC Cells

After the knockdown of PKCiota, the cell viability and colony formation ability of KYSE510 and KYSE450 cells were significantly decreased (Figure 2A–D). We further studied the roles of PKCiota silencing on ESCC cell ferroptosis. RSL3 was a well-used ferroptosis-induced agent that directly suppressed GPX4 [26]. Bright-field light microscopy showed that RSL3 induced cell death of KYSE510 and KYSE450 cells, and ferroptosis inhibitors (Liproxstatin-1, Lip-1, and Ferrostatin-1, Fer-1) but neither a necroptosis inhibitor (Necrostatin-1, Nec-1) or an apoptosis inhibitor (Z-VAD-FMK, ZVF) rescued RSL3-induced cell death (Figure 2E). A CCK-8 assay also confirmed that Lip-1 and Fer-1, but not ZVF and Nec-1, could significantly restore the decrease in cell viability induced by RSL3 (Figure 2F,G). The findings indicated that RSL3 caused ESCC cell ferroptosis. We also revealed that RSL3 downregulated the expression of PKCiota (Figure 2H and Appendix A). Another GPX4 inhibitor, FIN56, also downregulated the protein level of PKCiota in KYSE510 and KYSE450 cells. However, the xCT inhibitor Erastin did not affect the protein level of PKCiota in ESCC cells (Appendix A). Lip-1 and Fer-1 (not ZVF and Nec-1) significantly rescued the cell viability reduction caused by PKCiota silencing (Figure 2I,J). Silencing PKCiota also increased the MDA level in KYSE510 and KYSE450 cells (Figure 2K). All the above results suggested that the knockdown of PKCiota induced ESCC cell ferroptosis.

### 3.3. PKCiota Regulated USP14-Mediated Autophagic Degradation of GPX4 in ESCC Cells

Figure 3A and Appendix A show that silencing PKCiota reduced GPX4 protein expression but did not affect its mRNA level. After treatment with cycloheximide (CHX), the half-life of GPX4 in KYSE510 and KYSE450 cells was measured at indicated time points. The half-life of GPX4 was significantly shorter in PKCiota-silenced cells than in control cells (Figure 3B,C). The lysosome inhibitor Bafilomycin A1 (BafA1) absolutely suppressed GPX4 downregulation caused by PKCiota knockdown, and MG132 (a proteasome inhibitor) could partially recover GPX4 protein expression, which was reduced by PKCiota silencing (Figure 3D). Using the K63-linkage-specific polyubiquitin antibody, we found that silencing PKCiota increased K63-linked ubiquitination, which could lead to autophagic degradation (Figure 3E) [27,28]. These data indicated that PKCiota regulated GPX4 expression mainly via the autophagy–lysosome pathway.

Protein stability was frequently affected by the deubiquitinases, USP14 (a deubiquitinating enzyme) was overexpressed in ESCC, and the silencing of USP14 suppressed the tumorigenesis of ESCC [29,30,31]. Therefore, we evaluated whether PKCiota regulated GPX4 via regulating USP14. The co-IP method showed binding among PKCiota, GPX4, and USP14 in KYSE510 and KYSE450 cells (Figure 3F). Moreover, PLA verified the binding between PKCiota and GPX4 (Figure 3G). A co-IP assay further validated GPX4 bound to USP14 in ESCC cells (Figure 3H). The above results revealed that PKCiota formed a protein complex with GPX4 and USP14.

Then, we studied whether PKCiota regulated GPX4 expression by affecting USP14. Data indicated that the silence of PKCiota did not restrict the USP14 protein level but reduced the autophagy marker p62’s protein level, which further confirmed the regulatory role of PKCiota in autophagy (Figure 3I). Significantly, the USP14 inhibitor IU1 declined the GPX4 protein level, indicating that GPX4 expression was regulated by USP14 activity in ESCC cells (Figure 3J). A previous study reported that the phosphorylation of USP14 at Ser432 was necessary for activating its deubiquitinating activity [32]. Therefore, we analyzed whether PKCiota regulated the phosphorylation of USP14. Knockdown of PKCiota decreased the Ser phosphorylation of USP14 and enhanced the ubiquitination of GPX4 (Figure 3K). Recently, AKT was identified as the key enzyme for phosphorylating USP14 at Ser432 and regulating its deubiquitylation function. Our results showed that blocking AKT using MK2206 downregulated GPX4 and increased the K63-linked ubiquitination of GPX4 (Figure 3L,M). These findings suggested that PKCiota interacted with and activated USP14 deubiquitinating activity, then upregulated GPX4 expression via deubiquitinating GPX4 and blocking the autophagy–lysosome pathway, and finally suppressed the ferroptosis of esophageal cancer cells.

RSL3 treatment could decrease the protein level of GPX4 (Figure 3N). Importantly, we observed that the knockdown of PKCiota promoted the down-regulatory role of RSL3 on GPX4 expression (Figure 3O). The CCK-8 assay revealed that the silencing of PKCiota sensitized ESCC cells to RSL3-induced cell death (Figure 3P,Q). We further found that the knockdown of PKCiota and RSL3 treatment synergistically increased the cellular MDA level (Figure 3R).

### 3.4. PKCiota Was Regulated by a Positive Feedback Loop and Also Negatively Regulated by miR-145-5p in ESCC

Then, we evaluated whether there was a feedback regulatory loop between PKCiota and GPX4 or USP14. Interestingly, both the knockdown of USP14 and IU1 treatment reduced PKCiota protein levels (Figure 4A,B). And silencing GPX4 could also downregulate USP14 and PKCiota (Figure 4C). These results indicated that PKCiota formed a positive feedback loop with USP14 or GPX4.

In our previous study, miR-145-5p was downregulated in ESCC; meanwhile, its overexpression suppressed the proliferation of ESCC cells [33]. Therefore, we evaluated whether PKCiota was affected by miR-145-5p in ESCC. Interestingly, the miR-145-5p mimic reduced the mRNA and protein expression of PKCiota (Figure 4D–F). Like PKCiota knockdown, miR-145-5p mimics decreased GPX4 protein expression, not mRNA level (Figure 4G,H), and also did not regulate the protein level of USP14 (Figure 4I). miR-145-5p overexpression significantly decreased the colony formation abilities of KYSE510 and KYSE450 cells (Figure 4J). Importantly, miR-145-5p overexpression enhanced the down-regulatory role of RSL3 on GPX4 expression (Figure 4K). The mRNA level of PKCiota in the miR-145-5p mimic and RSL3 group was lower than in the miR-145-5p mimic-transfected group (Figure 4L). miR-145-5p overexpression sensitized ESCC cells to RSL3-induced cell death (Figure 4M,N). By analyzing the GEO datasets (GSE43732, GSE114110, and GSE145198), we revealed that miR-145-5p declined in ESCC (Appendix A).

### 3.5. Identification of Hub Genes Downstream of GPX4 in ESCC

Differentially expressed genes (DEGs) downstream of GPX4 were identified using the RNA-seq method. After GPX4 silencing, 282 downregulated and 83 upregulated genes were identified in KYSE510 cells (Figure 5A–C, Appendix A). The top five downregulated genes were AXDND1, PCDHB8, WAS, ITGBL1, and HOXB4, and the top five upregulated genes were CRYBB3, GRIP1, AL031777.3, ADCY10, and MYLK2 (Figure 5B,C, Appendix A). The top enriched terms identified in each GO category are shown in Figure 5D and Appendix A. The enriched pathways were the neuroactive ligand–receptor interaction and glycosphingolipid biosynthesis–lacto and neolacto series (Appendix A). The STRING website was applied to construct a PPI network based on DEGs. This network had 115 edges and 201 nodes (Figure 5E). We identified one module in this network with a setting k-score of more than 3 using the MCODE plugin in Cytoscape. The module included four genes (SOX2, FGF11, FGFR2, and BMP4), six edges, and four nodes (Figure 5F). We further identified hub genes in Cytoscape (cytoHubba plugin and two algorithms: Degree and MCC). Nine genes, namely SOX2, BMP4, FGFR2, SERPINA1, EGR1, WAS, VTN, CANX, and LUM, were identified by both the Degree and MCC methods (Figure 5G, Appendix A). Using qRT-PCR technology, we confirmed that the knockdown of GPX4 significantly downregulated the hub genes CANX, SOX2, and BMP4 (Figure 5H). Knockdown of GPX4 also reduced the protein levels of CANX and SOX2 in ESCC (Figure 5I). Downregulation of CANX was also detected in PKCiota-silenced cells (Figure 5J). Importantly, silencing CANX downregulated USP14 and GPX4 in KYSE510 and KYSE450 cells (Figure 5K).

### 3.6. Both Knockdown of PKCiota and miR-145-5p Overexpression Inhibited the Growth of ESCC In Vivo

In vivo experiments were performed to evaluate the effects of PKCiota and miR-145-5p on ESCC tumor growth. Tumor weight, together with volume, was detected weekly. Knockdown of PKCiota did not affect the body weight but significantly impeded tumor growth of ESCC cells (Figure 6A–D). The Western blotting method was applied to detect the expression of PKCiota and GPX4 in tumor tissues, and the GPX4 protein level was lower in PKCiota-silenced tumor tissues (Figure 6E). miR-145-5p overexpression did not affect the body weight but significantly repressed the tumor growth of KYSE510 cells (Figure 6F–I). The protein expression of GPX4 was lower in miR-145-5p-overexpressed tumor tissues (Figure 6J). We also revealed that silencing PKCiota with RSL3 treatment showed more substantial suppressive effects on tumor growth than the PKCiota knockdown alone group (Figure 6K–N). The combination of PKCiota silencing and RSL3 treatment further decreased the protein level of GPX4 more than only the PKCiota knockdown group (Figure 6O). These results confirmed the regulatory relationship between PKCiota and GPX4 in ESCC in vivo.

### 3.7. Expression and Prognostic Role of PKCiota and GPX4 in ESCC Samples

IHC experiments indicated that PKCiota and GPX4 were positively expressed in 45.8% and 31.3% of ESCC tissues (n = 48), respectively (Figure 7A,B). A significant correlation was found between positive staining of PKCiota and GPX4 (R = 0.6427, *p* < 0.0001, Appendix A). There was no significance between the co-expression of PKCiota and GPX4 and clinic parameters of ESCC patients (Appendix A). Notably, the patients with positive expression of both PKCiota and GPX4 showed shorter survival times than others (*p* = 0.0302, Figure 7C).

We further evaluated the prognostic role of the co-expression of PKCiota and USP14 in the GSE53625 dataset and found that the prognosis of patients with high PKCiota and USP14 and with high PKCiota or high USP14 was poorer than patients with low PKCiota and USP14 (Figure 7D).

Our findings suggested that PKCiota promoted the phosphorylation of USP14 at Ser sites and subsequentially reduced the ubiquitination of GPX4 via its Lys sites in ESCC. Therefore, the activation of USP14 and the mutation of Lys to other amino acids in GPX4 might activate the PKCiota/USP14/GPX4 axis, improve the protein stability of GPX4, and finally induce the resistance of ESCC cells to ferroptosis. By analyzing the cBioPortal database, we found that mutations to Ser in USP14 occurred in skin cutaneous melanoma, uterine endometrioid carcinoma, prostate adenocarcinoma, and ovarian cancer, and the mutation of Lys to other amino acids i GPX4 was detected in colorectal cancer (Figure 7E).

All the above findings suggested that PKCiota overexpression caused by *PRKCI* amplification and loss of miR-145-5p enhanced the phosphorylation of USP14 at Ser sites and subsequentially reduced the ubiquitination of GPX4, then improved the protein stability of GPX4 via suppressing the autophagy–lysosomal degradation pathway via forming a protein complex including PKCiota, USP14, and GPX4, and finally induced the resistance of ESCC cells to ferroptosis. Therefore, patients with loss of miR-145-5p, *PRKCI* amplification, mutation of Lys to other amino acids in GPX4, or activation of USP14 might be sensitive to GPX4-targeted ferroptosis-based therapy (Figure 7F).

## 4. Discussion

Our study indicated that PKCiota was downregulated during RSL3-induced ferroptosis, and silencing PKCiota especially sensitized ESCC cells to RSL3-induced ferroptosis. Up until now, the role of PKC during ferroptosis has been largely unclear. In rhabdomyosarcoma (RMS) cells, genetic knockdown of PKCα, Gö6976 (a PKCα and PKCβ selective inhibitor), and bisindolylmaleimide I (a pan-PKC inhibitor) suppressed Erastin-caused ferroptosis of RMS cells [14]. In Parkinson’s disease, PKCα activates the RAS/MEK signaling pathway and initiates ferroptosis [34]. These studies suggested that PKCα- and PKCβ-mediated ferroptosis participated in the progression of rhabdomyosarcoma and Parkinson’s disease. However, the role of other PKC members including PKCiota in ferroptosis is unclear. Our study reported that PKCiota was amplified and overexpressed in ESCC and caused the resistance of ESCC cells to ferroptosis.

Interestingly, we found that PKCiota interacted with GPX4 and increased its protein stability by suppressing the autophagy–lysosomal degradation pathway. GPX4 was the only enzyme to reduce cellular esterified phospholipid hydroperoxides and protect cells against ferroptosis [35]. GPX4 has been identified as the target for cancer therapy, especially for drug-resistant cancers [36,37,38]. Blocking the GPX4 and ferroptosis suppressor protein 1 (FSP1) pathways showed pronounced antitumor effects [39]. Bufotalin was reported to improve GPX4 ubiquitination and degradation, increase intracellular lipid ROS and Fe^2+^ levels, induce the ferroptosis of NSCLCs, and suppress tumor growth in vivo [40]. Sanguinarine (SAG) could reduce the protein stability of GPX4 via E3 ligase STUB1-induced ubiquitination and degradation of GPX4, then increase ROS, MDA, and Fe^2+^ levels, and finally cause the ferroptosis of NSCLCs and suppress tumor growth and metastasis in vivo [41]. FIN56 could promote autophagy-mediated GPX4 degradation and induce the ferroptosis of bladder cancer cells, and FIN56 and the mTOR inhibitor Torin 2 synergistically killed bladder cancer cells [42]. In Erastin-induced ferroptosis, acid sphingomyelinase (ASM) promoted the degradation of GPX4 [43]. The degradation of GPX4 was also regulated by its interaction with the proteins SMG9 and HSPA5 [44,45]. Although previous studies confirmed the regulation of GPX4 degradation by the proteasome or autophagy pathways [42,46], its detailed mechanisms in the ferroptosis of cancer cells, especially in ESCC cells, are unclear. Our results suggested that PKCiota was a critical factor in increasing the stability of GPX4 in ESCC.

Importantly, our results further showed that PKCiota, the deubiquitinase USP14, and GPX4 formed a protein complex; USP14 regulated the ubiquitination of GPX4; and PKCiota regulated the phosphorylation of USP14 in ESCC. We observed that blocking its activity by IU1 significantly decreased GPX4 expression. A previous study reported that the inhibition of deubiquitinases by using the palladium pyrithione complex (PdPT) promoted the degradation of GPX4 [47], but which deubiquitinase regulated GPX4 degradation was still unknown. Our results indicated that USP14 bound with GPX4, reduced its ubiquitination, and promoted the protein stability of GPX4 in ESCC.

USP14 has seven phosphorylation sites [48]. In ESCC, PKCiota did not affect USP14 expression but positively regulated the phosphorylation of USP14 at serine. A previous study reported that Akt could phosphorylate USP14 at Ser432 and activate its catalytic activity [32]. We previously reported that PKCiota could positively regulate AKT activation in ESCC [22]. Importantly, our findings indicated that blocking AKT using MK2206 reduced the protein levels of GPX4.

In our study, PKCiota overexpression was caused by its amplification, induced by loss of miR-145-5p, and regulated by GPX4 and USP14 through a positive feedback loop. Significantly, the knockdown of PKCiota and miR-145-5p overexpression sensitized ESCC cells to RSL3-induced ferroptosis. These results suggested that inducing ferroptosis might be used to treat ESCC in the future.

Mutation to Ser of USP14 and Lys mutation to other amino acids of GPX4 probably activate PKCiota/USP14/GPX4 axis, improve the protein stability of GPX4, and finally induce resistance of ESCC cells to ferroptosis. Very importantly, mutations to Ser in USP14 occurred in skin cutaneous melanoma, uterine endometrioid carcinoma, prostate adenocarcinoma, and ovarian cancer. In colorectal cancer, Lys mutation to other amino acids was detected in GPX4 [49,50]. These results indicated that the PKCiota/USP14/GPX4 axis might have participated in resistance to ferroptosis in these types of cancers.

Some problems still need to be solved in our future study. First, our results indicated that the knockdown of PKCiota diminished cell viability, lessened GPX4 levels, and enhanced cellular MDA levels; however, the effects of PKCiota overexpression on the cell viability, GPX4 level, and MDA level still need to be analyzed. Second, the exact Ser site of USP14 and Lys site of GPX4 which were regulated by PKCiota need to be identified.

## 5. Conclusions

Our findings suggested that PKCiota suppressed the ferroptosis of esophageal cancer cells by stopping the USP14-mediated autophagic degradation of GPX4. Targeting the PKCiota/USP14/GPX4 pathway might be developed to be a targeted therapeutic strategy for ESCC, especially for patients with loss of miR-145-5p, *PRKCI* amplification, mutation of Lys to other amino acids in GPX4, or activation of USP14.

## Figures and Tables

**Figure 1 antioxidants-13-00114-f001:**
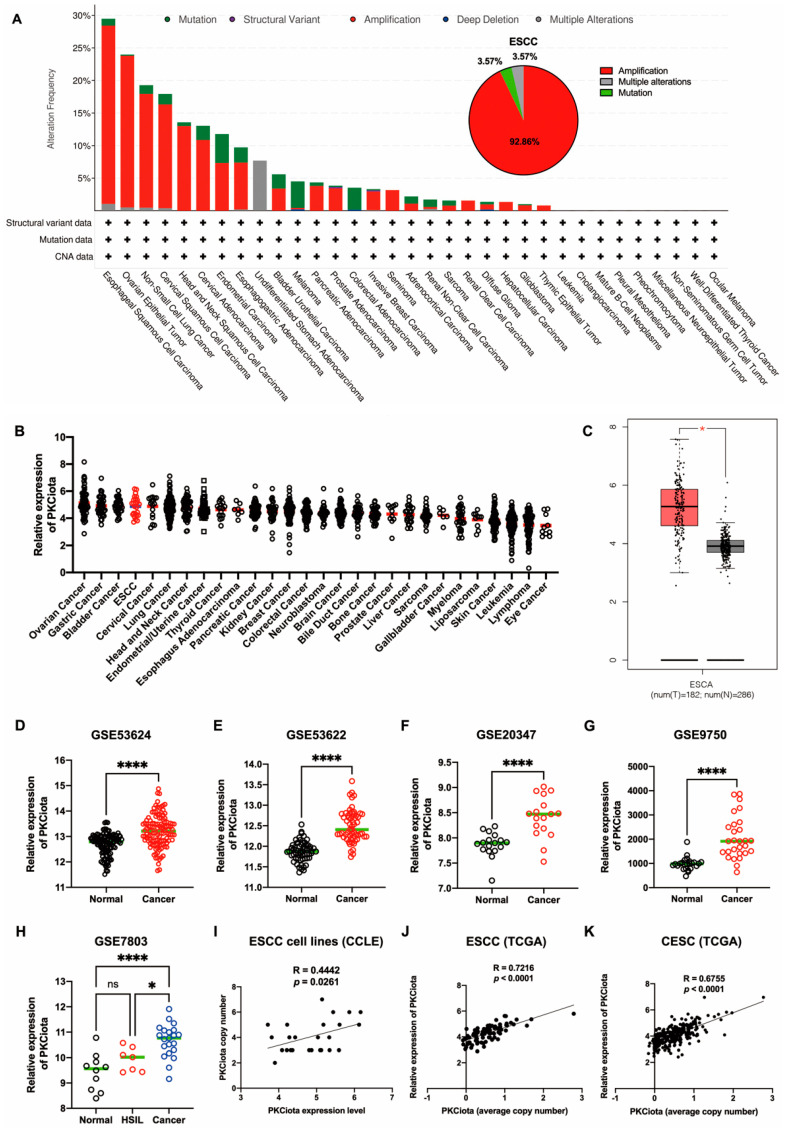
PKCiota was amplified and overexpressed in ESCC. Genetic changes in *PRKCI* in tumors (**A**), mRNA level of PKCiota in pan-cancer cell lines (**B**), and esophageal carcinoma (ESCA) tissues (**C**) were analyzed using cBioPortal, DepMap portal, and GEPIA databases. (**D**–**H**) Transcript levels of PKCiota in ESCC tissues and corresponding normal tissues (GSE53624, GSE53622, and GSE20347) and in CSCC tissues and corresponding normal tissues (GSE9750 and GSE7803). (**I**–**K**) The correlation between PKCiota copy number and its transcript level in cell lines, ESCC tissues, and CESC tissues was analyzed. ns: not significant; *: *p* < 0.05; ****: *p* < 0.0001.

**Figure 2 antioxidants-13-00114-f002:**
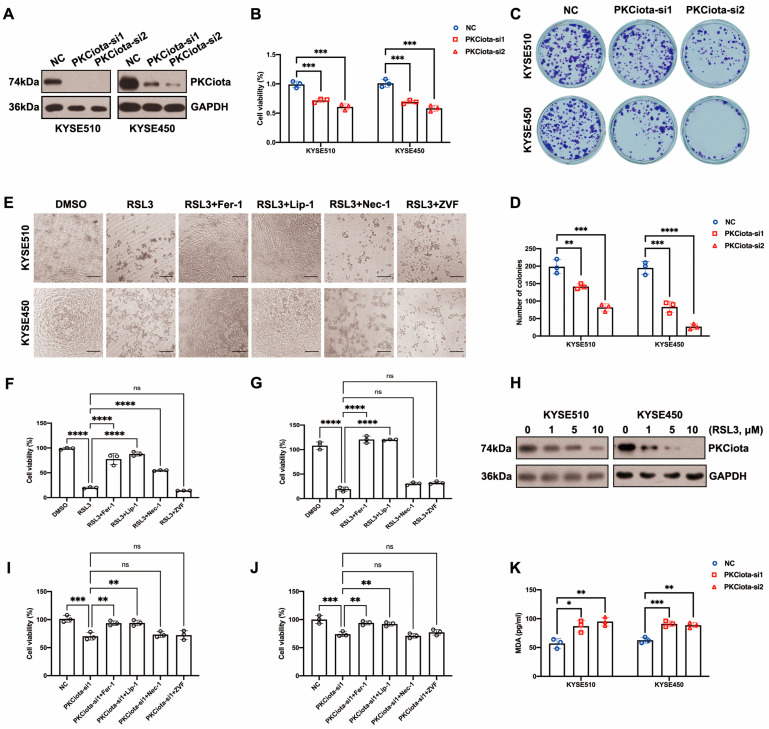
Knockdown of PKCiota reduced colony formation and cell viability, leading to ESCC cell ferroptosis. (**A**) PKCiota knockdown was confirmed using Western blotting assay at 48 h after transfection. (**B**–**D**) CCK-8 and colony formation assays were applied to evaluate cell viability and colony formation ability of ESCC cells after PKCiota knockdown. (**E**–**G**) Ferroptosis inhibitors (Lip-1/Fer-1, 2 μM) but not the apoptosis inhibitor (ZVF, 20 μM) and the necroptosis inhibitor (Nec-1, 10 μM) restored RSL3 (10 μM)-induced cell death. The images were captured, and the cell viability was measured 24 h after treatment (Scale bar = 50 μm). (**H**) Western blotting method was applied to detect PKCiota expression under RSL3 treatment for 48 h. (**I**,**J**) Fer-1/Lip-1 (2 μM), but not ZVF (20 μM) and Nec-1 (10 μM), restored the decline in cell viability caused by PKCiota silencing (detection at 48 h after transfection). (**K**) MDA level was detected after PKCiota siRNA transfection for 48 h in ESCC cells. ns: not significant; *: *p* < 0.05; **: *p* < 0.01; ***: *p* < 0.001; ****: *p* < 0.0001.

**Figure 3 antioxidants-13-00114-f003:**
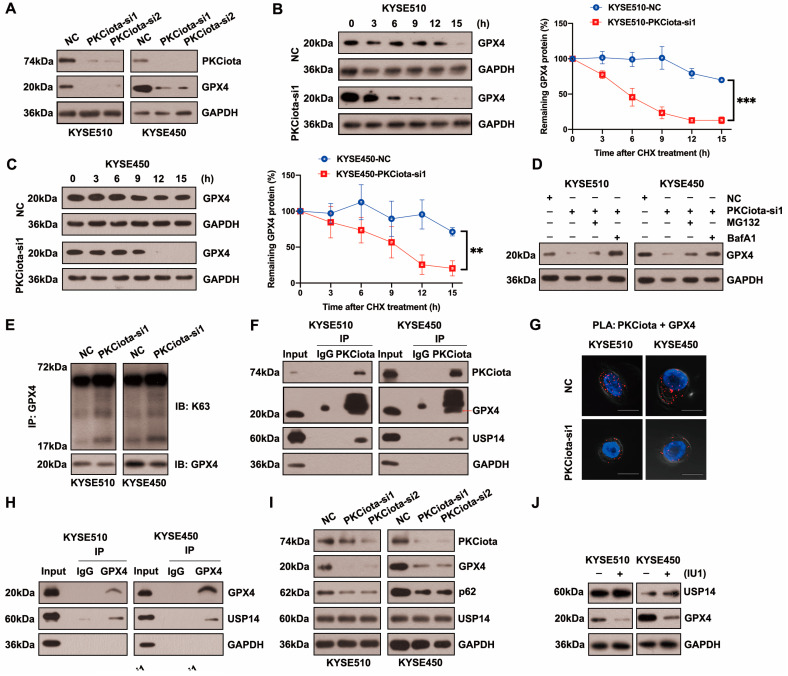
PKCiota regulated the USP14-mediated autophagic degradation of GPX4 in esophageal cancer cells. (**A**) Detection of GPX4 protein level at 48 h after PKCiota siRNA transfection by Western blotting method. (**B**,**C**) KYSE510 and KYSE450 cells after PKCiota knockdown were treated with CHX (10 μM), analyzed at 0, 3, 6, 9, 12, and 15 h, and immunoblotted for GPX4 and GAPDH. (**D**) Detection of GPX4 expression at 48 h after PKCiota siRNA transfection and MG132 (25 μM)- or BafA1 (100 nM)-added KYSE510 and KYSE450 cells by Western blotting method. (**E**) K63-linkage-specific polyubiquitin antibody was used to detect the K63-linked ubiquitination of GPX4. (**F**) Examination of the interaction among PKCiota, USP14, and GPX4 by co-IP assay. (**G**) Detection of the interaction between PKCiota and GPX4 by PLA technology (Scale bar = 10 μm). (**H**) The binding between GPX4 and USP14 was detected by a co-IP assay. (**I**) The effects of PKCiota silencing on the expression of GPX4, p62, and USP14 were analyzed through Western blotting assay at 48 h after PKCiota siRNA transfection. (**J**) The effects of IU1 (100 μM, 48 h) on GPX4 expression were detected. (**K**) The effects of PKCiota knockdown on phosphorylation at Ser sites of USP14 and ubiquitination of GPX4 at 48 h after PKCiota siRNA transfection were evaluated through Co-IP and Western blotting methods. (**L**) Protein expression of p-AKT, AKT, USP14, and GPX4 after MK2206 (10 μM) treatment was analyzed via Western blotting technology. (**M**) K63-linkage-specific polyubiquitin antibody was used to detect the K63-linked ubiquitination of GPX4. (**N**) Protein level of GPX4 under RSL3 treatment (48 h). (**O**) Effects of PKCiota knockdown and RSL3 treatment on expression of GPX4. (**P**,**Q**) Cell viability of KYSE510 and KYSE450 cells after PKCiota knockdown and RSL3 treatment (CCK-8 assay). (**R**) Cellular MDA was measured after PKCiota knockdown and RSL3 treatment. ns: not significant; **: *p* < 0.01; ***: *p* < 0.001; ****: *p* < 0.0001.

**Figure 4 antioxidants-13-00114-f004:**
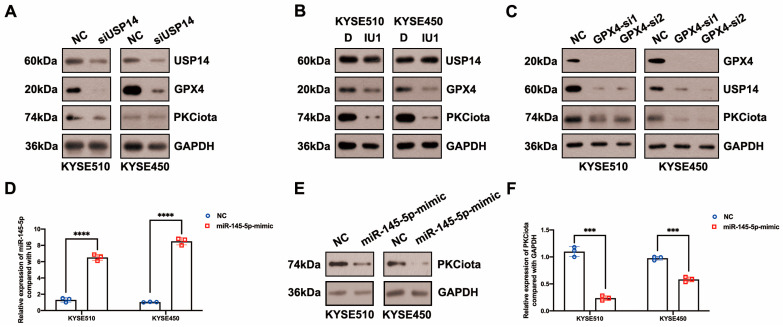
PKCiota was regulated by a positive feedback loop and also negatively regulated by miR-145-5p in ESCC. (**A**,**B**) Effects of USP14 knockdown (48 h after USP14 siRNA transfection) or IU1 (100 μM, 48 h) treatment on the protein levels of PKCiota, GPX4, and USP14 were evaluated by Western blotting. (**C**) Effects of GPX4 knockdown (48 h after GPX4 siRNA transfection) on the expression of PKCiota, USP14, and GPX4 were evaluated by the Western blotting method. (**D**) miR-145-5p expression after transfection (48 h) was detected using qRT-PCR. (**E**–**I**) Effects of miR-145-5p mimic transfection (48 h) on PKCiota, GPX4, and USP14 were evaluated. (**J**) Effects of miR-145-5p mimic transfection on the colony formation abilities of ESCC cells. (**K**) Effects of miR-145-5p mimic transfection together with RSL3 treatment on expression of GPX4. (**L**) The transcript level of PKCiota under miR-145-5p mimic transfection and RSL3 treatment. (**M**,**N**) Cell viability of KYSE510 and KYSE450 cells under RSL3 treatment together with miR-145-5p mimic transfection. ns: not significant; *: *p* < 0.05; **: *p* < 0.01; ***: *p* < 0.001; ****: *p* < 0.0001.

**Figure 5 antioxidants-13-00114-f005:**
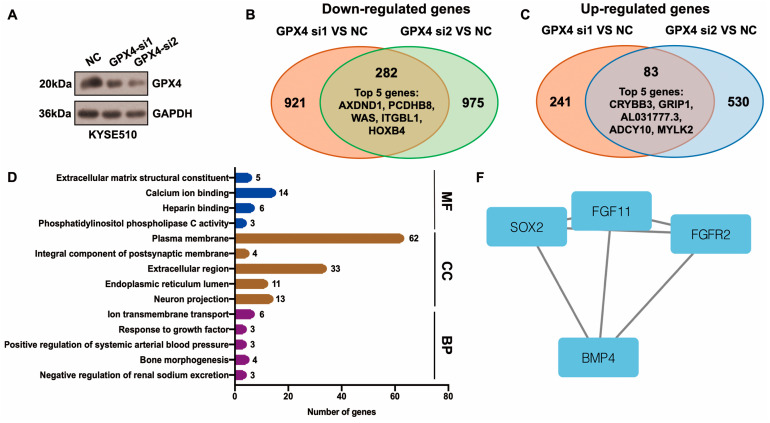
Identification of hub genes downstream of GPX4 in ESCC. (**A**) GPX4 knockdown was confirmed using Western blotting assay 48 h after GPX4 siRNA transfection. (**B**,**C**) Venn plot showed the overlapped downregulated and upregulated genes after GPX4 knockdown (si1 and si2). (**D**) GO analysis of DEGs after GPX4 knockdown in KYSE510 cells. (**E**) The PPI network was constructed based on DEGs using the STRING website. (**F**) Module in the PPI was identified using the MCODE plugin in Cytoscape. (**G**) Hub genes were selected through the cytoHubba plugin in Cytoscap (MCC algorithm). (**H**) Transcript levels of CANX, SOX2, and BMP4 after GPX4 knockdown. (**I**) Protein expression of CANX and SOX2 after GPX4 knockdown. (**J**) Protein expression of CANX after PKCiota knockdown. (**K**) Protein expression of USP14 and GPX4 after CANX silence. *: *p* < 0.05; **: *p* < 0.01; ***: *p* < 0.001; ****: *p* < 0.0001.

**Figure 6 antioxidants-13-00114-f006:**
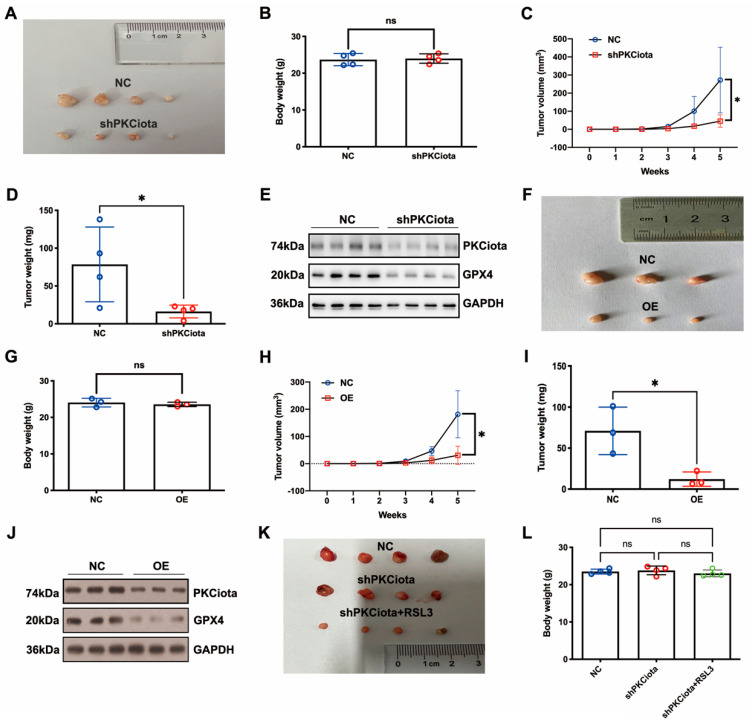
Knockdown of PKCiota and miR-145-5p overexpression suppressed the growth of ESCC in vivo. PKCiota-silenced (**A**–**E**), miR-145-5p-overexpressed (**F**–**J**), and PKCiota-silenced together with RSL3-treated (**K**–**O**) KYSE510 cells were inoculated into mice subcutaneously. Body weight, tumor weight, and volume were measured. Protein expression of GPX4 and PKCiota in tumor tissues was detected. ns: not significant; *: *p* < 0.05; **: *p* < 0.01; ***: *p* < 0.001.

**Figure 7 antioxidants-13-00114-f007:**
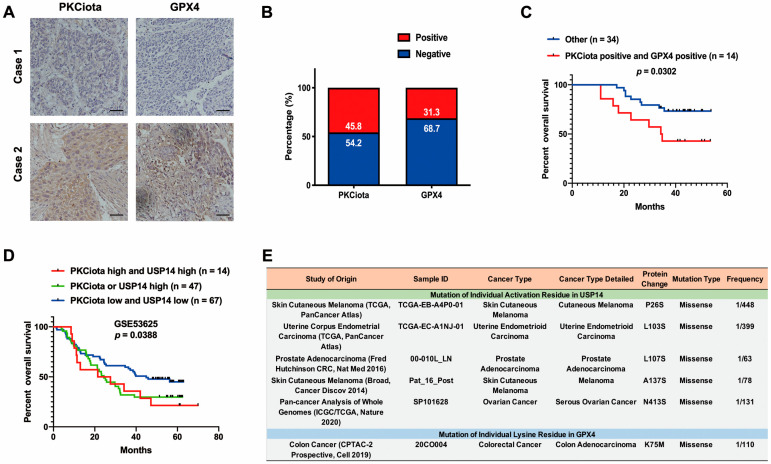
Expression and prognostic role of PKCiota and GPX4 in ESCC samples. (**A**) Representative IHC images of PKCiota and GPX4 in tumor samples. (**B**) Positive rates of PKCiota and GPX4 in tumors (Scale bar = 50 μm). (**C**) The Kaplan–Meier curve shows the association between lower survival rate and co-expression of PKCiota and GPX4 in ESCC. (**D**) Kaplan–Meier curve showing the association between survival time of ESCC patients and positive expression of PKCiota and USP14 in the GSE53625 dataset. (**E**) The cBioPortal database was used to analyze the mutation to Ser of USP14 and Lys mutation to other amino acids in GPX4 in pan-cancer. (**F**) Schematic of mechanisms underlying PKCiota-induced resistance of ESCC cells to ferroptosis.

## Data Availability

The data that support the findings of this study are available from the corresponding author upon reasonable request.

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
