# Peer review of "PKCiota Inhibits the Ferroptosis of Esophageal Cancer Cells via Suppressing USP14-Mediated Autophagic Degradation of GPX4"

_antioxidants, 2024, doi:10.3390/antiox13010114_

Round 1

Reviewer 1 Report

Comments and Suggestions for Authors

Esophageal squamous cell carcinoma (ESCC) is an aggressive type of cancer with poorly defined apoptosis mechanism. This research aims to understand how PKCiota, influences the resistance of ESCC cells to ferroptosis, a form of cell death characterized by the iron-dependent accumulation of lipid peroxides. The study found that high levels of PKCiota in ESCC cells reduced their vulnerability to ferroptosis presumably by PKCiota mediated phosphorylation events. PKCiota interacted with the proteasome associated DUB, USP14, leading to increased stability of GPX4 via reduced ubiquitination and resistance to ferroptosis. miR-145-5p also contributed to PKCiota's effects. Silencing PKCiota and increasing miR-145-5p levels in these cells made them more susceptible to ferroptosis. The findings suggest that targeting the PKCiota/USP14/GPX4 pathway could be a potential approach for treating ESCC

Comments:

Some of the figures are difficult to interpret. In particular those where siRNA is used. It would be much more useful to the reader if the siRNA target was listed on the figure instead of just si1, si2 etc. 

The authors state that ‘’Knockdown of PKCiota decreased the Ser phosphorylation of USP14, and meanwhile enhanced the ubiquitination of GPX4 (Figure 3K)’’ However the blot is not entirely convincing since there is very little change in the KYSE510 cell line. Furthermore, I am not sure if this method is entirely suited for detection of posttranslational modifications (Fig 3E, 3K, 3M). Were the cells lysed using a denaturing lysis buffer? The authors used Roche kit but there is no info available on whether this is done under denaturing conditions. There is always a danger of pulling down/loosing interacting proteins when normal IP conditions are used. Lastly I would encourage the authors to list detailed methods in supplementary material instead of citing previous publications (the co-IP protocol cites to a publication that cites another publication before the method can be read). 

The animal experiments outlined in fig 6 suggest a role for PKCiota as a anti tumor target. However the tumor weight and volume at the end of the experiment is very small. Is there a reason for terminating the experiment at the 5 week timepoint as opposed to allowing a longer growth period to allow assessment of more larger tumors. 

Minor comments: 

Esophageal is in bold in abstract.

Abbreviation definition of ESCC should also be included in the first sentence of introduction. 

Comments on the Quality of English Language

Overall the manuscript is written in an easy understandable manner. Some small copy editing to improve grammar and word flow would be beneficial.

Author Response

Comments and Suggestions for Authors

Esophageal squamous cell carcinoma (ESCC) is an aggressive type of cancer with poorly defined apoptosis mechanism. This research aims to understand how PKCiota, influences the resistance of ESCC cells to ferroptosis, a form of cell death characterized by the iron-dependent accumulation of lipid peroxides. The study found that high levels of PKCiota in ESCC cells reduced their vulnerability to ferroptosis presumably by PKCiota mediated phosphorylation events. PKCiota interacted with the proteasome associated DUB, USP14, leading to increased stability of GPX4 via reduced ubiquitination and resistance to ferroptosis. miR-145-5p also contributed to PKCiota's effects. Silencing PKCiota and increasing miR-145-5p levels in these cells made them more susceptible to ferroptosis. The findings suggest that targeting the PKCiota/USP14/GPX4 pathway could be a potential approach for treating ESCC

Comments:

Some of the figures are difficult to interpret. In particular those where siRNA is used. It would be much more useful to the reader if the siRNA target was listed on the figure instead of just si1, si2 etc.

Thanks for your valuable suggestions, we carefully revised our figures, especially adding the siRNA target.

The authors state that ‘’Knockdown of PKCiota decreased the Ser phosphorylation of USP14, and meanwhile enhanced the ubiquitination of GPX4 (Figure 3K)’’ However the blot is not entirely convincing since there is very little change in the KYSE510 cell line. Furthermore, I am not sure if this method is entirely suited for detection of posttranslational modifications (Fig 3E, 3K, 3M). Were the cells lysed using a denaturing lysis buffer? The authors used Roche kit but there is no info available on whether this is done under denaturing conditions. There is always a danger of pulling down/loosing interacting proteins when normal IP conditions are used. Lastly I would encourage the authors to list detailed methods in supplementary material instead of citing previous publications (the co-IP protocol cites to a publication that cites another publication before the method can be read).

Thanks for your valuable suggestions, because of the lack of commercial antibodies against phosphorylated USP14 (especially Ser432 site or all Ser sites), after referring to the literature of “Xu et al. Phosphorylation and activation of ubiquitin-specific protease-14 by Akt regulates the ubiquitin-proteasome system. Elife. 2015; 4: e10510”, the present method was used to detect the phosphorylation of USP14 at Ser sites (Fig 3K). After referring to the literature of “Zhang et al. Extracellular fibrinogen-binding protein released by intracellular Staphylococcus aureus suppresses host immunity by targeting TRAF3. Nat Commun. 2022; 13: 5493”, the present method was used to detect the K63-linked ubiquitination of GPX4 (Fig 3E and 3M). Thanks for your valuable suggestions, the cells in the co-IP assay were lysed using an IP lysis buffer (non-denaturing), and we also revised and provided more detailed information on the methods.

The animal experiments outlined in fig 6 suggest a role for PKCiota as a anti tumor target. However the tumor weight and volume at the end of the experiment is very small. Is there a reason for terminating the experiment at the 5 week timepoint as opposed to allowing a longer growth period to allow assessment of more larger tumors.

Thanks for your valuable suggestions, we will optimize our research procedure and terminate the experiment after a longer growth period to allow the assessment of larger tumors in our future study.

Minor comments:

Esophageal is in bold in abstract.

Thanks for your valuable suggestions, we corrected this mistake.

Abbreviation definition of ESCC should also be included in the first sentence of introduction.

Thanks for your valuable suggestions, we added the abbreviation definition of ESCC in the first sentence of the introduction.

Reviewer 2 Report

Comments and Suggestions for Authors

the authors do a good job in investigating a relatively novel pathways in ESCC. Overall the study is very comprehensive and the data is relevant and solid.

The main questions and needs reside in expanding the methods descriptions - for example, there is no indication on how imaging was performed, antibody dilutions, parts 2.6 and 2.8 will benefit from brief descriptions instead of just adding a citation, for the animal studies there needs to be a mention on power analysis and how the number of mice was chosen per group.  It is also important to clarify the replicates performed in each experiment (i.e., is it technical triplicates or biological triplicates (if the second is true then how many technical replicates were done?). all images miss the scale bar, and the brightfield images are low quality (artifacts present - i.e., striping, illumination unevenness). 

Comments on the Quality of English Language

The language is overall reasonable but it would need some editing as there are several parts that are not quite correct

Author Response

Comments and Suggestions for Authors

the authors do a good job in investigating a relatively novel pathways in ESCC. Overall the study is very comprehensive and the data is relevant and solid.

The main questions and needs reside in expanding the methods descriptions - for example, there is no indication on how imaging was performed, antibody dilutions, parts 2.6 and 2.8 will benefit from brief descriptions instead of just adding a citation, for the animal studies there needs to be a mention on power analysis and how the number of mice was chosen per group.  It is also important to clarify the replicates performed in each experiment (i.e., is it technical triplicates or biological triplicates (if the second is true then how many technical replicates were done?). all images miss the scale bar, and the brightfield images are low quality (artifacts present - i.e., striping, illumination unevenness).

Thanks for your valuable suggestions, we carefully revised the method descriptions and added more detailed information. We provide information on how imaging was performed, antibody dilutions, and biological replicates of each experiment. We improved the brightfield images' contrast.

Reviewer 3 Report

Comments and Suggestions for Authors

In this MS, authors suggested that amplified and overexpressed PKCiota induced resistance of ESCC cells to ferroptosis via suppressing USP14-mediated autophagic degradation of GPX4. Generally it was well organized and written. Minor comments were suggested.

1.      Add animal study approval number

2.      How about effect of PKCiota overexpression on the viability, GPX and MDA level

3.      Check English expression and careless spelling

Comments on the Quality of English Language

Check careless flaws and spelling

Author Response

Comments on the Quality of English Language

The language is overall reasonable but it would need some editing as there are several parts that are not quite correct

Thanks for your valuable suggestions, we carefully revised the English expressions of our manuscript.

Comments and Suggestions for Authors

In this MS, authors suggested that amplified and overexpressed PKCiota induced resistance of ESCC cells to ferroptosis via suppressing USP14-mediated autophagic degradation of GPX4. Generally it was well organized and written. Minor comments were suggested.

1.Add animal study approval number

Because the animal study approval of our study hasn’t an approval number. Therefore, we have provided the approval file from the Animal Ethics Committee of Kunming University of Science and Technology to the editor at the time of manuscript submission. And we also provided the information on the date of approval in the manusript.

  1. How about effect of PKCiota overexpression on the viability, GPX and MDA level

Thanks for your valuable suggestions, we discussed the limitations of our study in the section of discussion. The effect of PKCiota overexpression on cell viability, GPX4 expression, and MDA level will be studied in our future study.

  1. Check English expression and careless spelling.

According to your suggestion, we carefully revised the English expressions of our manuscript and corrected all the mistakes.

Comments on the Quality of English Language

Check careless flaws and spelling

According to the reviewer’s suggestion, we carefully revised the English expressions of our manuscript and corrected all the mistakes.

Round 2

Reviewer 1 Report

Comments and Suggestions for Authors

The manuscript is much improved and suitable for publication

Reviewer 2 Report

Comments and Suggestions for Authors

we thank the authors for addressing the reviewer's concerns

Reviewer 3 Report

Comments and Suggestions for Authors

Thanks. Much improved.